# Corn Starch-Chitosan Nanocomposite Film Containing Nettle Essential Oil Nanoemulsions and Starch Nanocrystals: Optimization and Characterization

**DOI:** 10.3390/polym13132113

**Published:** 2021-06-28

**Authors:** Fatemeh Kalateh-Seifari, Shima Yousefi, Hamed Ahari, Seyed Hedayat Hosseini

**Affiliations:** 1Department of Agriculture and Food Science, Science and Research Branch, Islamic Azad University, Tehran 476714171, Iran; f_kalateh00@yahoo.com (F.K.-S.); Shyousefi81@gmail.com (S.Y.); 2Department of Food Science and Technology, Shahid Beheshti University of Medical Science, Tehran 1983969411, Iran; Seyed.Hedayat@gmail.com

**Keywords:** chitosan-starch composite, nettle essential oil, starch nanocrystals, optimization

## Abstract

In the current study, nanocomposite films were produced based on corn starch:chitosan (CS:CH) biopolymers and the films were reinforced with nettle essential oil nanoemulsions (NEONEs) and starch nanocrystals (SNCs) to improve their physicochemical and mechanical properties. CS: CH at 70:30, 50:50, and 30:70 (*w*/*w*) ratios; SNCs at 2, 4, and 6% (*w*/*w*), and NEONEs at 0.5, 1, and 1.5% (*w*/*w*) were selected as variables. Then the various physical and mechanical attributes of chitosan-starch blended film containing SNCs and NEONEs were optimized using response surface methodology. The desirability function technique for the second-order polynomial models revealed that the following results could be achieved as the optimized treatment: water solubility of 51.56%; water absorption capacity of 128.75%; surface color of L (89.60), a (0.96), and b (1.90); water vapor permeability of 0.335 g/s Pa m, oxygen permeability of 2.60 cm^3^ μm/m^2^ d kPa; thickness of 154.41 µm, elongation at break of 53.54%; and tensile strength of 0.20 MPa at CS:CH of 38:62, SNC of 6.0%, and NEONEs of 0.41%. The nanocomposite film obtained can be employed as a novel biofunctional film with boosted physical mechanical and physical characteristics for food packaging applications.

## 1. Introduction

In recent years, the development of bio-based degradable materials for food packaging applications has been considered significant due to the various negative impacts of petroleum-based synthetic materials on ecosystems, health, sustainability, groundwater, etc. [1]. Bio-based polymeric materials are commonly classified into three main groups including lipid-based, protein-based, and carbohydrate-based raw materials [2]. Most of them are cost-effective, renewable, abundant, and highly accessible sources derived from food, agriculture, and seafood byproducts [3]. Such bio-based materials possess some inherent process-ability drawbacks in comparison with their synthetic counterparts, limiting their further implementation on an industrial scale [4]. Employing some physical and/or chemical modification strategies can turn them to promising alternatives to petroleum-based packaging materials. Introducing nanostructures (nanoparticles, nanotubes, nanofibers, etc.), crosslinking agents, plasticizers, biopolymers blending to obtain biocomposites, antimicrobial agents, and bioactives are among highly recommended approaches to boost the applicability of biomaterials for food packaging uses [5].

Chitosan is one of the valuable biopolymers for food packaging and preservation applications due to several benefits such as biodegradability, biocompatibility, non-toxicity, abundance, and antimicrobial properties [6]. Chitosan polysaccharide is the soluble form of chitin and can be obtained by partial deacetylation of chitin biopolymers. Chitosan showed suitable film-forming capacity with excellent antimicrobial, barrier, and antioxidant attributes. Blending of chitosan with other biopolymers is an effective way to improve its film-forming capacity and functionality [7]. In this regard, blending chitosan with polysaccharide-based biopolymers demonstrated a couple of advantages comprising low cost, suitable stability, proper sealability, and high availability [8]. It has been proven that chitosan-starch blend films can fabricate biodegradable packaging materials with improved mechanical, thermal, barrier, and biofunctional characteristics. Starch biopolymer is able to fabricate transparent, flexible, and thin layer films, while it has some drawbacks such as weak barrier and shear strength features [9].

In addition to biopolymer blending, loading nanostructured ingredients into the biopolymer matrix is suggested as a novel method to strengthen their operational characteristics. Nanostructured biopolymers are known as bionanocomposites, regarded as potential competitors instead of non-degradable plastic materials. Starch nanocrystals (SNCs) are natural eco-friendly nanostructures possessing outstanding physicochemical attributes to reinforce packaging film matrices [10]. SNCs are basically manufactured by hydrolysis of amorphous parts of starch biopolymer chains to segregate crystalline regions in order to achieve SNCs with particle size lower than 100 nm. Such nanostructures have different applications in the generation of Pickering emulsions, and the improvement of physicochemical and barrier attributes of packaging films and coatings [11].

Development of active food packaging systems is another category of interest during the last years. Nanobiocomposites with antimicrobial attributes attracted great attention as a safe method to extend the shelf life of food products through the inactivation of microbial growth [12]. Essential oils are phenolic compounds derived from plant secondary metabolites capable of numerous emerging antimicrobial properties. Essential oils are naturally occurring concentrated hydrophobic volatile compounds extracted from most of the herbal plants with several therapeutic and health-promoting attributes. Tea tree essential oil [13], cinnamon essential oil [14], oregano and thyme essential oils [15], and ginger essential oils [16] are some examples of incorporation of essential oils into the chitosan, zein, soy, and starch biocomposites, respectively. Nettle (*Urtica dioica* L.) essential oil (NEO) contains various bioactive compounds including flavonoids, tannins, carotenoids, vitamins, etc. which possess various antiviral, antifungal, and antioxidant activities [17]. Incorporation of essential oils into the nano-vehicles can preserve their availability and stability during the processing, transportation and shelf life once loaded to nanobiocomposite packaging films. Therefore, encapsulation of NEO by nanoemulsion systems can produce a stable functional compound to fabricate active food packaging films [18].

To the best of our knowledge, co-incorporation of SNCs and NEO nanoemulsions (NEONEs) into the chitosan-starch blend film has not been investigated so far. The core purpose of the current study was to optimize the various physical and mechanical attributes of chitosan-starch blend film containing SNCs and NEONEs using response surface methodology–Box-Behnken design (RSM-BBD). The chosen treatments then exposed to the various physicochemical and mechanical tests.

## 2. Materials and Methods

### 2.1. Chemicals and Reagents

Fresh aerial parts of nettle were harvested from a forest area of Northern Iran (Chabokasr, Guilan, Iran). Sunflower oil was obtained from Tabiat Sabz Co. (Tehran, Iran). Corn starch, chitosan (MW: 300 kDa, 80–90% deacetylation), acetic acid, sorbitan monooleate (Tween 80), potassium sorbate, ammonium thiocyanate, glycerol, sulfuric acid, and sodium sulfate were bought from Merck Co. (Darmstadt, Germany). Other chemicals and reagents were of analytical grade.

### 2.2. Methods

#### 2.2.1. Starch Nanocrystals (SNCs) Preparation and Characterization

The SNCs were obtained according to a method developed by Garcia et al. [19] with slight modification. Briefly, starch granules were hydrolyzed by a sulfuric acid solution (98%) under agitation (100 rpm) at 40 °C for 5 days. The achieved suspension was then centrifuged at 1000 rpm at 4 °C for 30 min and the resulted supernatant pellet were collected and washed several times by milli-Q water to remove the acid residuals. Afterwards, the pellets were diluted in water and subjected to an ultrasonication homogenizer (Model 3000, Biologic Inc., Manassas, VA, USA), at 24 kHz for 15 min, to obtain well-distributed SNCs. Finally, the sonicated suspension was spray dried by using a lab-scale spray drier (Buchi B290, Flawil, Switzerland) to achieve SNCs powder.

The morphology of SNCs was tested using a scanning electron microscope (SEM) (Model XL20, Phillips, Eindhoven, The Netherlands) under the acceleration voltage of 26 kV. The X-ray diffraction (XRD) pattern of the fabricated SNCs was also examined by using a diffractometer instrument (X-Pert Pro, PANalytical, Amsterdam, The Netherlands) at diffraction angle range of 0–100° and scan step of 0.1° equipped with Cu anode at 20 mA and 40 kV. The mean particle size of the synthesized SNCs was determined by using a Zetasizer apparatus (Malvern Inc., Malvern, UK) at 25 °C and scattering angle of 90° according to dynamic light scattering (DLS) technique.

#### 2.2.2. Preparation of Nettle Essential Oil (NEO) and the Nanoemulsions (NEs)

Preparation of NEOs and NEONEs were performed using a method described by Gharibzahedi et al. [17]. In view of that, fresh nettle leaves first partially sun-dried and then freeze-died (Model FD-4, Pishtaz Co., Tehran, Iran) to remove further moisture and avoid bioactive compound’s lost. Hydrodistillation was conducted by adding 500 g of the dried leaves into 2.0 L of distilled water in a flask which was connected to a Clevenger apparatus for 5 h at boiling point. The obtained NEO desiccated over anhydrous sodium sulfate and kept at 4 °C until use.

The oil in water nanoemulsion was manufactured by aqueous to oil phase ratio of 90 to 10 wt% using a high speed ultrahomogenizer (Ultraturrax T25, Stauffen, Germany) for 3 min at 12,000 rpm at ambient temperature. It should be noted that 10% of the aqueous phase was containing Tween 80 as surfactant and the aqueous phase was dispersed in an acetate solution buffer (10 Mm at pH 4.0). Oil phase was also composed of 2.5% of NEO and 7.5% of sunflower oil. The prepared nanoemulsions (NEONEs) were kept in glass flasks enclosed by parafilm at ambient temperature until use.

#### 2.2.3. Identification of NEO Composition

Identification of the chemical composition of NEO was carried out by gas chromatography (GC 7890N, AGILENT and MS 5975C). The specifications and temperature settings of the device were as follows. The chromatographic device used was of the AGILENT type with a 30 m long column, the inner diameter was 0.25 mm and the layer thickness was 0.25 µm. The apparatus temperature increased from 60 °C to 250 °C at 5 °C and was kept at 250 °C for 1 min. Helium gas with a flow rate of 1 mL/min was used. The injection volume was 0.5 μL and the identification of the compounds was standardized using standard compounds and the available information in the device library. In order to determine the percentage of identified compounds, gas chromatography (GC) connected to the mass spectrograph. Each volatile component was identified by the relevant GC Kovats indices and the correspond mass spectra in mass spectra library (Adams, NIST, and Wiley).

#### 2.2.4. Composite Film Preparation Using Casting Method

Nanocomposites were fabricated using solvent casting method as depicted by Mehdizadeh et al. [20]. Chitosan films were prepared by addition of 2% (*w*/*v*) chitosan biopolymer into an aqueous solution containing 1% (*v*/*v*) acetic acid under stirring at 300 rpm and 70 °C overnight on a hotplate. The prepared solution then filtrated through a Whatman filter paper No. 1 to wipe off the undissolved substances. In like manner, the corn starch films were prepared by dissolving 3.5% (*w*/*v*) starch in bi-distilled water at 300 rpm for 30 min at 90 °C until starch gelatinization happened, and then the suspension cooled down up to 40 °C. The corn starch–chitosan (CS–CH) composite films were also formulated using the 2% and 3.5% solutions of chitosan and corn starch solutions, respectively, in various ratios. Glycerol was also included in the composite films at the level of 25% (*w*/*w*) of the total dry matter of the composites as plasticizer. The NEONEs produced were subjected to 0.2% surfactant prior to load into the composite matrices. All prepared solutions were then thoroughly homogenized at 7000 rpm for 3 min. The uniformed solutions were finally casted in Teflon plates and then peeled off after 72 h at ambient temperature. The obtained composite films were exposed to relative humidity (RH) of 50% for 72 h until use.

#### 2.2.5. Film Characterization

##### Water Solubility (WS)

Water solubility of the composite films was determined according to Mehdizadeh et al. [20] with brief modification. The composite films cut into 2 cm × 2 cm dimensions and desiccated at 60 °C overnight in an oven. The desiccated pieces were then inserted into the boiling tubes for 1 h prior to filtration step. The second desiccation step was carried out for 24 h again at 60 °C. The final weights of composite films were measured and their water solubility contents were calculated as follows (Equation (1)):
(1)$$\mathrm{Water}\;\mathrm{solubility}\;\left(\%\right)=\frac{\mathrm{Initial}\;\mathrm{dry}\;\mathrm{weight}-\mathrm{Final}\;\mathrm{dry}\;\mathrm{weight}}{\mathrm{Initial}\;\mathrm{dry}\;\mathrm{weight}}\times 100$$

##### Water Absorption Capacity (WAC)

The water absorption capacity (WAC) of the composite films were measured according to the method describe by Sadegh-Hassani and Nafchi [21]. Briefly, the composite films were desiccated in phosphorus pentoxide for 7 days and then cut into 2 cm × 2 cm pieces and then soaked in 100 mL of bi-distilled water for 30 min at ambient temperature. The swollen composite films came out from water and weighted and the amount of the absorbed water by film matrix is considered as WAC.

##### Film Thickness

The composite film thickness was determined using a digital micrometer (Model, DC 516, Reed Instruments Ltd., Wilmington, NC 28412, USA). The average of 10 measured points was reported as the mean film thickness.

##### Surface Color

The surface color of films was calculated by a colorimeter device (Model CR-300, Minolta Co., Tokyo, Japan). The CIELab color measurement system was employed to determine lightness (L*), redness (a*) and yellowness (b*) values of composite films. The samples were placed on a standard white sheet of apparatus with the following values as default: L* = 97.61, a* = −0.17, and b* = 0.49. The total colour change (ΔE) was also calculated as follows:ΔE = [(ΔL*)^2^ + (Δa*)^2^ + (Δb*)^2^]^0.5^(2)
where, L_0_, a_0,_ and b_0_ are colour values of the control sample and the L_1_, a_1_ and b_1_ are values for other samples.

##### Elongation at Break (EB) and Tensile Strength (TS)

In order to prepare the samples for mechanical tests the composites first cut into 10 cm × 0.5 cm pieces and placed in a desiccator with 50% RH at ambient temperature for 24 h to reach moisture equilibrium point. An Instron tensile tester (Model 5542, Canton, MA, USA) with cross head speed of 10 mm/min and initial grip separation of 50 mm was used to determine mechanical attributes. The elongation at break (EB) and tensile strength (TS) values were calculated based on the force extension curves according to ASTM D882 [22]. EB (%) is measured by the initial and final length of composite films at their rupture point and TS (MPa) values can be calculated by the following equation: (3)$$\mathrm{TS}=\frac{\mathrm{Maximum}\;\mathrm{Force}}{\left(\mathrm{Film}\;\mathrm{thickness}\right)\times \left(\mathrm{Film}\;\mathrm{width}\right)}$$

##### Water Vapor Permeability (WVP)

The glass cups with 3.0 cm internal diameter and 3.5 cm height were used to determine the water vapor permeability (WVP) of films. The glass cups were loaded with calcium chloride desiccant to generate RH of 0%. Afterwards, the cup surfaces were covered by composite film samples followed by cup sealing using molten paraffin. The cups then moved to a desiccator with RH of 100% and weighted over 7 days with a weighting interval of 24 h. The weight loss over the experiment time was plotted to attain the curve slope. Finally, the water vapor transition rate (WVTR) and WVP were calculated as follows [23]:(4)$$\mathrm{WVTR}=\frac{\;\mathrm{Curve}\;\mathrm{slope}}{\mathrm{Film}\;\mathrm{area}}$$
(5)$$\mathrm{WVP}=\frac{\mathrm{WVTR}\;\times \mathrm{Thickness}}{\mathrm{Pressure}\;\mathrm{difference}}$$

It should be noted that the RH variances between two sides of the composite forms a vapor pressure of about 3200 Pa at ambient temperature.

##### Oxygen Permeability (OP)

The oxygen permeability (OP) of the films was measured by a permeance testing apparatus (Model GDP-C, Munchen, Germany) according to Chakravartula et al. [24]. Once more the composite films cut into 2 cm × 2 cm dimensions and placed between the apparatus chambers. The upper chamber was filled with food-grade oxygen at atmospheric pressure, and gas permeation was measured by determining the pressure increase in the lower chamber. The experiments performed at gas stream of 100 cm^3^/min, RH of 0%, and film area of 0.650 cm^2^.

#### 2.2.6. Experimental Design and Statistical Analysis

Experimental design, data analysis and model building were employed by using software Design Expert (Version 11.0, Stat-Ease Inc., Minneapolis, MN, USA). The effect of three independent variables, namely X_1_ (CS:CH: 70:30, 50:50, and 30:70 *w*/*w* ratios which shown as 7, 5 and 3, respectively) and X_2_ (SNCs: 2, 4, and 6% *w*/*w*), and X_3_ (NEONEs: 0.5, 1, and 1.5% *w*/*w*) on solubility (Y_1_), WAC (Y_2_), TM (Y_3_), L* (Y_4_), a* (Y_5_), b* (Y_6_), TS (Y_7_), EB (Y_8_), WVP (Y_9_), and OP (Y_10_) was evaluated using the RSM-BBD as presented in Table 1. The range and center point values of three independent variables were based on the results of preliminary experiments. The data were fitted to a second order polynomial equation as a function of the dependent variables. The coefficients of the polynomial were represented by b_0_ (constant term), b_1_, b_2_ and b_3_ (linear effects), b_11_, b_22_ and b_33_ (quadratic effects), and b_12_, b_13_ and b_23_ (interaction effects), and the second-order polynomial equation was obtained as follows:(6)$$y=b_{0}+b_{1}x_{1}+b_{2}x_{2}+b_{3}x_{3}+b_{12}x_{1}x_{2}+b_{13}x_{1}x_{3}+b_{23}x_{2}x_{3}+b_{11}x_{1}^{2}+b_{22}x_{2}^{2}+b_{3}x_{3}^{2}$$

The significant terms (*p* < 0.05) were chosen for final equations. The best fitting mathematical model was selected according to the comparisons of several statistical parameters including the coefficient of variation (CV), the multiple correlation coefficient (R^2^), the adjusted multiple correlation coefficient (adj-R^2^), the prediction error sum of squares (PRESS), and adequate precision (AP) proved by the used software. The use of response surface graphs and surface contour plots also provided a visual aid in examining the effect of variables on each factor and in determining the optimal levels to fabricate nanocomposite films.

## 3. Results and Discussion

### 3.1. The Phenolic Compounds of Nettle Essential Oils

The GC results showed that 2,4-dichlorophenol with 47.7% and pentachlorophenol with 23.5% were the most abundant components of NEO (Table 2). The chemical composition of NEO in the present study was different from the chemical composition reported for this essential oil by Gharibzahedi et al. [18], which can be attributed to the factors of growing location, climate and growing season while the chemical composition of this essential oil was consistent with the findings of Lahigi et al. [25].

### 3.2. Characterization of SNCs

Figure 1 demonstrates the morphology, XRD-pattern, and particle size distribution of the synthesized SNCs. As can be seen in Figure 1a, with an electric current intensity of 20 kV and a magnification of 20.00 KX, the average particle size shown is about 102 nm, and particles seem to have a uniform distribution although some clustering/aggregations were observed. Different materials have various diffraction patterns due to their dissimilar arrangement and atomic orders. In the same material with crystalline phases, the diffraction pattern is also different. Therefore, by studying the angle at which the XRD peaks are formed (Figure 1b) and the relative intensity of each peak, the type of materials and their phase can be qualitatively identified; amorphous materials do not form specific peaks, while crystalline materials that have a regular structure create specific peaks at certain angles Jarzebski et al. [26] suggested that for the DLS analysis, the intensity of scattered light by particles is of utmost importance for nanoemulsion characterization. The reason could be due to the presence of some large particles that strongly affect the average particle’s diameter. The XRD test is a common test for measuring the distribution of nanoparticles in a structure and measuring the crystallinity of nanoparticles and nanocomposites. Figure 1b shows the results of this test for SNCs prepared by acid hydrolysis. The synthesized SNC exhibited two sharp diffraction peaks at 2θ = ~18° and ~23°. Similar results were obtained from previous studies where there are similar diffraction peaks for different types of starch nanocrystals [27,28]. For instance, Xu et al. [28] used XRD analysis to demonstrate that the starches derived from corn, barley and tapioca have a typical crystalline structure with diffraction peaks at about 17.0, 17.61 and 22.81. The particle size distribution and mean particle size of the prepared Starch nanoparticles (SNPs) were also monitored by the DLS method (Figure 1c). As is obvious, a slender size distribution with polydispersity index (PDI) of 0.192 was observed for SNCs with a mean diameter of 114 ± 3.7 nm. The size of SNCs in both methods were almost close to each other, while the freeze dried sample was relatively smaller than the hydrated sample in DLS analysis, probably due to the swelling of SNCs in aqueous media and/or the occurrence of aggregation between individual particles dispersed in water [29].

### 3.3. Characterization of Nanocomposite Films

Analyses of variance (ANOVAs) helped to produce the quadratic models for response surfaces, which elucidated the fitness, accuracy, and significance of the models in addition to the effects of interaction results and individual variables on the responses (Table 3).

#### 3.3.1. Solubility

The effect of X_1_ (CS:CH ratio), X_2_ (SNCs concentration) and X3 (NEONEs concentration) on Solubility (Y_1_) were evaluated through regression analyses (Table 1). The resulting quadratic equation was determined as follow:(7)$$Y_{\mathrm{solubilty}}=59.39+14.18X_{1}-1.70X_{2}-5.94X_{3}+0.5X_{1}X_{2}+0.625X_{2}X_{3}-1.92X_{1}^{2}+1.97X_{2}^{2}-1.12X_{3}^{2}$$

The correlation coefficients R^2^ and adjusted R^2^ were evaluated with the latter reflecting an adjustment for number of model parameters relative to the number of points in the study. R^2^ for Y1 was 0.999% indicated a strong relationship between experimental and predicted values (Figure 2a). However, the adjusted R^2^ for Y_1_ (0.9976%) also reflects the influence of independent variables. A non-significant lack of fit (*p* > 0.05; shows the suitability of models to accurately predict the variations. The R^2^ value was 0.999 for the solubility. Table 2 reveals that the solubility of the produced film was significantly influenced by the linear (*p* < 0.0001), quadratic (*p* < 0.01; *p* < 0.05) and interaction (*p* < 0.05) effects of independent variables studied. Based on the regression coefficients and F ratio (Table 2), the importance of the independent variables on solubility could be ranked in the following order: CS:CH > NEO-NEs > SNCs concentration. However, the largest effect on solubility was the positive linear term of CS:CH content followed by the negative linear term of NEONEs and quadratic term of SNCs concentration. The three-dimensional response surfaces were based on Equation (7) and helped to understand the interactive and main effects of independent variables (Figure 2b). It can be seen that when starch/chitosan content was increased from 3% to 7% (*w*/*w*), the solubility increased in a parabolic manner. In contrast, increasing the SNCs and also NEONEs concentrations led to a significant decrease in the aforementioned physical characteristics.

In general, solubility is an important factor in determining the possibility of using a film as a package, which affects the film’s resistance to water, especially in humid environments. According to the research, the high solubility of the film in water, especially for packaging fresh and frozen products, is a disadvantage, while the high standard is also an advantage for some packaging applications. The film’s solubility depends on water diffusion, amino and carboxylic groups, and the separation and hydration of hydrogen and ionic bonds [30]. The results showed that the solubility of the film increased with increasing starch-to-chitosan ratio. Due to the severe hydrophilicity of the chitosan, the reduction in the amount of chitosan resulted in an increase in the solubility of the film. Also, the increase in NEONE and SNCs reduced the solubility of the film. This could be due to the interference of essential oil molecules with the polymer structure and hydrogen bonds, which can cause interference in the formation of bonds between the polymer molecules. In fact, the addition of NEO to the chitosan film led to the formation of covalent bonds and reduced accessibility between the functional group of amine and hydroxide of chitosan chain and essential oil which led to limited water-polysaccharide reactions by hydrogen bonds. Hence, solubility of chitosan film was decreased [31]. The non-polar compounds in NEO have a repulsive effect on water molecules. Therefore, cross linking of NEO in the chitosan film matrix can generate strong intra-molecular interactions of NEO components with amide, amine and hydroxyl groups of chitosan. Therefore, higher intra-molecular interactions in chitosan lead to a matrix with low affinity to water [32,33,34]. This phenomenon coincides with the obtained results about the addition of rosemary [32], cinnamon [33] and carvacrol [34] into chitosan films.

#### 3.3.2. Water Absorption Capacity (WAC)

WAC was best described by the regression Equation (8) after sequential omission of the non-significant factors (Table 2). The model could explain 97.88% of the behavior of WAC, which also had a non-significant lack of fit (*p* = 0.0518). In order to predict quadratic polynomial model, multiple regression coefficients was made by least squares technique and regarding coefficient significance, the following model was proposed:(8)$$Y_{2}=174.60+62.74X_{1}-5.56{\;X}_{2}-11.08X_{3}+5.14{\;X}_{1}X_{3}-6.951.97X_{2}^{2}\;$$

Data indicate that the WAC directly related to the SNCs and NEO concentration as well as CS:CH ratio. Moreover, the mutual interaction between these parameters also was significant and had the largest effect on WAC. Figure 3a depict the interaction between CS:CH and SNCs concentration and also interaction between SNCs and NEO-NEs concentration on WAC of the produced film. Figure 3b clearly shows that WBC can be enhanced by increasing the SC:CH ratio. Due to the hydrophilic nature of starch, pure starch film shows high moisture absorption, but the addition of SNCs and NEONE increased with increasing CS: CH ratio. Similar results have been reported on the effect of NEONE on film water absorption [32,35].

#### 3.3.3. Thickness Measurement (TM)

The thickness of films in this study was best described by the regression Equation (9) and regarding to coefficient significance, the following model was proposed:(9)$$Y_{\mathrm{TM}}=0.1521+0.0045X_{1}+0.0033X_{2}+0.0044{\;X}_{3}+0.001X_{1}X_{3}-0.0012X_{2}X_{3}+0.0033X_{1}^{2}-0.0032X_{3}^{2}$$

A non-significant lack of fit (*p* > 0.05; Table 2) shows the suitability of models to accurately predict the variations. The R^2^ value was 0.9843 for the thickness (Table 2). The $R_{\mathrm{adj}}^{2}$ (0.9641) is a modification of R^2^ that adjusts for the number of explanatory terms in a model. Table 2 reveals that the thickness of the produced film was significantly influenced by the linear (*p* < 0.0001), quadratic (*p* < 0.01) and interaction (*p* < 0.05) effects of independent variables studied. Based on the regression coefficients and F ratio, the importance of the independent variables on thickness could be ranked in the following order: CS:CH > NEONEs > SNCs concentration. However, the largest effect on thickness was the positive linear term of CS:CH content followed by the positive linear term of NEONEs concentration.

Figure 4c shows the interaction between CS:CH and nano essential concentration and also interaction between SNCs and NEONEs concentration on thickness of the produced film. From the three-dimensional and perturbation graphs (Figure 4c), it can be seen that when starch/chitosan content was increased from 3% to 7% (*w*/*w*), the thickness increased in a parabolic manner. Moreover, increasing the SNCs and also NEONEs concentrations led to a significant increase in the mentioned physical characteristic. The results showed that the increase in NEONEs and SNCs concentration, as well as the higher starch to chitosan ratio, produced thicker packaging films. This increase can be attributed to the confinement of micro-droplets of essential oil in Chitosan’s film [36]. The results of Shojaee-Aliabadi et al. [37] study in kappa-carrageenan film containing safflower essential oil were consistent with the results of this study.

#### 3.3.4. Color

After regression analysis the following model was proposed for L* (lightness), a* (redness) and b* (yellowness) parameters, respectively:(10)$$Y_{L*}=87.98-3.89X_{1}-1.03{\;X}_{3}-0.4825{\;X}_{1}X_{3}-1.53X_{1}^{2}$$
(11)$$Y_{a*}=1.01+0.0180X_{1}+0.0120X_{2}+0.0420X_{3}-0.01X_{1}X_{3}+0.1114X_{1}^{2}-0.0786X_{2}^{2}+0.1014X_{3}^{2}$$
(12)$$Y_{b*}=1.79-0.2340X_{1}+0.2320X_{3}$$

Table 2 reveals that the lightness of the produced film was significantly influenced by the linear (*p* < 0.0001), quadratic (*p* < 0.001) and interaction (*p* < 0.05) effects of independent variables studied. Based on the regression coefficients and F ratio (Table 4), the importance of the independent variables on lightness could be ranked in the following order: CS:CH > NEONEs > SNCs concentration. However, the largest effect on lightness was the positive linear term of CS:CH content followed by the negative linear term of NEONEs concentration. From the three-dimensional and perturbation graphs (Figure 5), it can be seen that when CS:CH content was increased from 3% to 7% (*w*/*w*), the lightness increased in a parabolic manner. Moreover, increasing the SNCs and also NEONEs concentrations led to a significant increase in the mentioned physical characteristic. In the case of essential oils, they increase with increasing essential oil and then slightly decreased. According to Figure 6, when CS:CH content was increased, the redness increased in a parabolic manner. Moreover, increasing the SNCs and also NEONE concentrations led to a significant increase in the aforementioned physical characteristic. In the case of essential oils, they increased with increasing essential oil and then slightly decreased. With increasing CS:CH content, the yellowness decreased whereas, increasing of NEONEs led to decrease of yellowness (Figure 7). Therefore, the nano-active films containing encapsulated NE were more yellow than control film and free NE-loaded films. As shown in Table 2, addition of both free and encapsulated NE increased color difference (ΔE) and led to darker films. The color of the packaging film is one of the most important factors influencing the consumer. Indicators of turbidity, color and film clarity are important factors for the overall appearance and acceptance of the consumer. In general, adding different ingredients, if colored, causes color changes in the films, and these changes are more evident in the addition of extracts that are more colored than essential oils that are usually colorless or slightly yellowish. High concentration SNCs probably due to the formation of large aggregates with the high water-holding potential and strong interactions with other hydrophilic components in coating solutions can significantly reduce the lightness value [38]. Similar results have been obtained by using green tea extract in chitosan film [39]. A study by Ojagh et al. [33] showed that the color change of the films was completely influenced by the type and amount of essential oil used, even at low concentrations so that increasing the concentration of various essential oils from 0.5% to 1% reduces the whiteness index of films.

#### 3.3.5. Mechanical Properties

The regression coefficients were calculated according to the multiple regression coefficients and a polynomial regression model equation was fitted as follows:(13)$$Y_{3}=0.2459+0.099X_{1}+0.0841X_{2}+0.0393{\;X}_{3}+0.0412X_{1}X_{2}-0.0798X_{1}^{2}$$

The data showed that the tensile strength was directly related to the CS:CH ratio, SNCs and NEONEs concentration. Moreover, the mutual interaction between these parameters also was significant on TS. The model could explain 96.51% of the behavior of tensile strength, which also had a non-significant lack of fit (*p* = 0.0038). Based on the regression coefficients and F ratio (Table 5), the importance of the independent variables on tensile strength could be ranked in the following order: CS:CH > SNCs > NEONEs concentration. However, the largest effect on tensile strength was the positive linear term of CS:CH content followed by the positive linear term of SNCs concentration. Figure 8 (a and b) depicts the interaction between CS:CH and NEONEs concentration and also the interaction between SNCs and NEONEs concentration on tensile strength of the produced film. From the three-dimension and perturbation graphs (Figure 8), it can be seen that when CS:CH content was increased from 3% to 7% (*w*/*w*), the tensile strength increased in a parabolic manner. Moreover, increasing the SNCs and also NEONEs concentrations led to a significant increase in the aforementioned physical characteristics.

The regression coefficients were calculated according to the multiple regression coefficients and a polynomial regression model equation was fitted as follows:(14)$$Y_{\mathrm{EB}}=36.64-6.60X_{1}+6.62X_{2}+5.23X_{3}-3.32X_{1}X_{2}-2.71X_{1}X_{3}-11.73X_{1}^{2}+8.69X_{2}^{2}+4.65X_{3}^{2}$$

The data showed that the EB was directly related to the CS:CH ratio, SNCs and NEONEs concentration. Moreover, the mutual interaction between these parameters also was significant on EB. The model could explain 97.08% of the behavior of EB, which also had a non-significant lack of fit (*p* = 0.0038). Table 5 discloses that EB of the produced film was significantly influenced by the linear, quadratic and interaction effects of independent variables studied. Based on the regression coefficients and F ratio the importance of the independent variables on EB could be ranked in the following order: SNCs > CS:CH > NEONEs concentration. However, the largest effect on EB was the positive linear term of SNCs content followed by the positive linear term of CS:CH concentration. Figure 9a,b depicts the interaction between CS:CH and NEONEs concentration and also interaction between SNCs and NEONEs concentration on EB of the produced film. From the three-dimension and perturbation graphs (Figure 9), it can be seen that when CS:CH content was increased from 3% to 7% (*w*/*w*), the EB increased in a parabolic manner. Moreover, increasing the SNCs and also NEONEs concentrations led to a significant increase in the aforementioned physical characteristics.

One of the most important features of food packaging materials is their mechanical properties. Optimizing the mechanical properties of edible films is important for the following aspects: the high mechanical strength of the film prevents mechanical damage such as perforation as a result of stress and thus retains its inhibitory properties against gases and moisture. The high flexibility of the film makes it compatible with the shape of the food without breaking it and it can be easily used as a cover. The film’s high mechanical strength protects the food inside from stress during transport. Starch produces films with low tensile strength and high flexibility that are not suitable for use in food packaging at all. That’s why improving the mechanical properties of starch films is one of the topics that has been the focus of many researchers [40,41,42]. Using chitosan and essential oils are one of the strategies to overcome the strength of starch films. Film elongation at break is related to flexibility and the elongation capacity of the materials. The results showed that increasing the EB of films requires changing the ratio of chitosan to starch and the ratio of other additives. Flexibility increase as a function of other additives content may be related to structural changes in the starch network because the matrix becomes less dense and, under tension, the movements of the polymer chains are facilitated. Chitosan acted as a reinforcing agent in starch-based biodegradable films. Thus, higher content of chitosan can render the films stiffer. As a result, a decrease in EB values was observed. Similar results were reported by Pinotti et al. [43] who indicated the reduction in methylcellulose film flexibility with increasing chitosan concentration. According to Sun et al. [44] when the chitosan/starch films are used as a film, the problem of film rupture often occurred. Hence, the elongation at break of such films must be increased. The mechanical properties of chitosan and starch at different ratios and with concentration gradients were compared by Xu et al. [28]. The results showed that the tensile strength and elongation of the films increase first and then decrease as starch content increases. The maximum EB of the film can reach 60%. Therefore, by changing the ratio of chitosan to starch, the elongation at break of the films can be increased. Tuhin et al. [45] showed that the mechanical properties of the chitosan/starch films had a great relationship with the hydrogen bond strength, crystallinity and the internal structure of the films.

Based on previous researches, different essential oils have shown different effects on mechanical properties of chitosan film. For example, thyme, clove and cinnamon essential oils increase EB and TS [46], cinnamon and ginger led to an improvement of 328% and 111% of the EB with cinnamon and ginger, respectively [47]. Also, citronella and cedar-wood increased TS [48] whereas five essential oils (ginger, rosemary, sage, tea tree and thyme) increased the TS of chitosan film. Generally, essential oils act as plasticizers and increase the flexibility of polymer chains. The presence of essential oils in films can lead to the formation of weak network structures. As a result, TS may decrease and EB may increase. The differences between mechanical properties of different essential oil may be attributed to the type of chitosan (molecular weight and solvent) used and the particular interactions with the essential oil components which, in turn, are affected by relative humidity, the presence of surfactants, temperature, etc. [49], caracole, grape seed extract also decrease tensile strength [50].

#### 3.3.6. Water Vapor Permeability

The reduced quadratic regression model in terms of coded factors for the water vapor permeability (Y_7_) was developed as follows:(15)$$Y_{7}=0.3649+0.0752X_{1}-0.0189X_{2}-0.0137X_{3}+0.0150X_{1}X_{3}+0.02X_{2}^{2}$$

A non-significant lack of fit (*p* > 0.05) shows the suitability of models to accurately predict the variations. The R^2^ value was 0.9788 for the WVP. The $R_{\mathrm{adj}}^{2}$ (0.9516) is a modification of R^2^ that adjusts for the number of explanatory terms in a model. Table 5 reveals that the WVP of the produced film was significantly influenced by the linear (*p* < 0.0001, *p* < 0.05), quadratic (*p* < 0.05) and interaction (*p* < 0.05) effects of independent variables studied. Based on the regression coefficients and F ratio, the importance of the independent variables on WVP could be ranked in the following order: CS:CH > SNCs > NEONEs concentration. However, the largest effect on WVP was the positive linear term of CS:CH content followed by the negative linear term of SNCs and quadratic term of SNCs concentration. Figure 10a,b shows the interaction between CS:CH and NEONEs concentration on WVP of the produced film. From the three-dimensional, counter and perturbation graphs (Figure 10), it can be seen that when CS:CH content was increased from 3% to 7% (*w*/*w*), the WVP increased in a parabolic manner. In contrast, increasing the SNCs and also NEONEs concentrations led to a significant decrease in the aforementioned physical characteristics.

Water vapor permeability properties are one of the most important factors in the application of food packaging. Depending on the type of food, food films should be able to prevent or at least reduce the transfer of moisture between the food and the atmosphere around the food. Various approaches have been used to reduce water vapor permeability, including the use of cross-linking materials, the production of two- or multi-layered films, and the production of chitosan composites with moisture-resistant biological materials and other polymers such as starch and waxes [50]. Another approach is to use hydrophobic compounds including fatty acids and essential oils and extracts. In addition to improving antimicrobial and antioxidant properties, essential oils are hydrophobic compounds, so their use in chitosan reduces the film’s hydrophilicity and reduces WVP [49,51].

The results showed that permeability decreased with increasing essential oil content. Because the transfer of water vapor occurs through the hydrophilic parts of the film, therefore the water vapor permeability depends on the ratio between hydrophobic and hydrophilic compounds in the film [52]. In addition, the WVP index of food films strongly affected by chemical structure, morphology of the film, the conditions of preparation and the type and concentration of additives. The phenolic compounds in the extracts, which are located in the chitosan matrix, form a series of hydrogen and covalent bonds with the active chitosan groups. Hydrogen and covalent bonds between the chitosan network and phenolic compounds reduce the ability of hydrogen groups to form hydrogen bonds with water and ultimately reduce the film’s tendency to water [13].

Similar results were observed by Bonilla et al. [51] in the case of chitosan-based film containing basil and thyme essential oils. Perdones et al. [52] also reported reduction in WVP in case of chitosan film containing 3% lemon essential oil. However, in some other studies, adding essential oils increased WVP [53,54]. This difference in the results of various researches can be due to the difference in the chemical structure of different essential oils. According to the results of the present study, due to the low level of permeability in the starch film, the amount of permeability decreased with the combination of starch and chitosan, which is considered a desirable state. According to the results conducted Xu et al. [28] on the composition of starch and chitosan also confirmed the findings of the present study. Due to their fine structure, nanoparticles can be easily filled in porous film matrix, making it difficult to lose moisture or water. When a nanoparticle is present in a polymer matrix, a water molecule must move on a more complex path than the pure polymer [55].

#### 3.3.7. Oxygen Permeability (OP)

The generalized polynomial model proposed for predicting the response variables is given as:(16)$$Y_{8}=3.40+1.28X_{1}-0.295X_{3}+0.2775X_{1}^{2}$$

A non-significant lack of fit (*p* > 0.05; Table 6) shows the suitability of models to accurately predict the variations. The R^2^ value was 0.9927 for the OP (Table 6). The $R_{\mathrm{adj}}^{2}$ (0.9834) is a modification of R^2^ that adjusts for the number of explanatory terms in a model. Table 6 reveals that the OP of the produced film was significantly influenced by the linear (*p* < 0.0001, *p* < 0.001), quadratic (*p* < 0.05) effects of independent variables studied. Based on the regression coefficients and F ratio, the importance of the independent variables on OP could be ranked in the following order: CS:CH > NEONEs concentration. However, the largest effect on OP was the positive linear term of CS:CH content followed by the negative linear term of NEONEs. From the three-dimensional, counter and perturbation graphs (Figure 11), it can be seen that when CS:CH content was increased from 3% to 7% (*w*/*w*), the OP increased in a parabolic manner. In contrast, increasing the NEONEs concentrations led to a significant decrease in the aforementioned physical characteristics. The oxygen diffusion from air into the package should be avoided because the product quality can be modified during prolonged storage due to the oxidation, which leads to changes in organoleptic properties and decreased nutritional value. The oxygen permeation through a material is calculated as the weight or volume of oxygen which is able to pass through a known area of a packaging material in a given time. Generally, gas permeability depends on film microstructure (holes or discontinuities in the polymer structure), thickness, the solubility of gases in the corresponding material, and the level of arrangement of the polymer chains [56]. The results showed that with the addition of essential oils, the oxygen permeability was reduced. Most likely, the essential oils have created a structure with fewer spaces between the networks by filling the voids created in the large network with the linear structure of the chitosan. There is limited information available on this subject, but in general, adding essential oils improves gas permeability. The results showed that as the SNCs concentration increased, the rate of OP decreased significantly. This result was in agreement with the result reported by González et al. [57]. They observed that SNCs could be considered as impermeable barriers to the movement of oxygen molecules. Moreover, the oxygen barrier properties are influenced by the thickness of the chitosan layer applied. Hence, the thicker the chitosan layer, the lower the OP.

### 3.4. Optimization and Verification of Models

The RSM evaluated the effects and interactions of the CS:CS, NEO-NEs and SNCs to optimize film properties (Table 7). The desirability function technique for the second-order polynomial models revealed that the following results could be achieved as the optimized treatment: the WS of 51.56%; WAC of 128.75%; surface color of L* (89.60), a* (0.96), and b* (1.90); WVP of 0.335 g/s Pa m, OP of 2.60 cm^3^ μm/m^2^ d kPa; TM of 154.41 µm, EB of 53.54%; and TS of 0.20 MPa at CS:CH of 38:62, SNC of 6.0%, and NEONEs of 0.41%. Under the suggested optimal conditions, the corresponding experimental values for the WS, WAC, surface color of L*, a*, b*, WVP, OP, TM, EB and TS of the produced nanocomposite film are presented in Table 7, respectively. Validation step showed a good agreement between the predicted and experimental data. Therefore, response surface optimization properly predicted the optimal conditions.

## 4. Conclusions

The core objective of the current study was to examine strengthening mediators, namely NEONEs and SNCs, to develop a modified chitosan–starch nanocomposite film for food packaging uses. The prepared NEONEs were rich in phenolic compounds and showed an average particle size of about 102 nm coupled with a heterogeneous scattering pattern, in addition to homogenous particles. The particle size distribution and mean particle size of the prepared SNPs showed that a slender size distribution with polydispersity index of 0.192 was observed for SNCs with a mean diameter of 114 nm. The optimization approach was efficient enough to select an appropriate combination of film ingredients (CS:CH of 38:62, SNC of 6.0%, and NEONEs of 0.41%) in order to achieve a biodegradable nanocomposite film with excellent physical and mechanical attributes. Finally, it can be concluded that chitosan–starch nanocomposite films can be proposed as a promising natural packaging material with suitable physical and mechanical characteristics as well as improved water, gas and oxygen permeation for food packaging and biomedical applications. The fabricated films could be used as suitable antimicrobial films due to the presence of NEONEs.

## Figures and Tables

**Figure 1 polymers-13-02113-f001:**
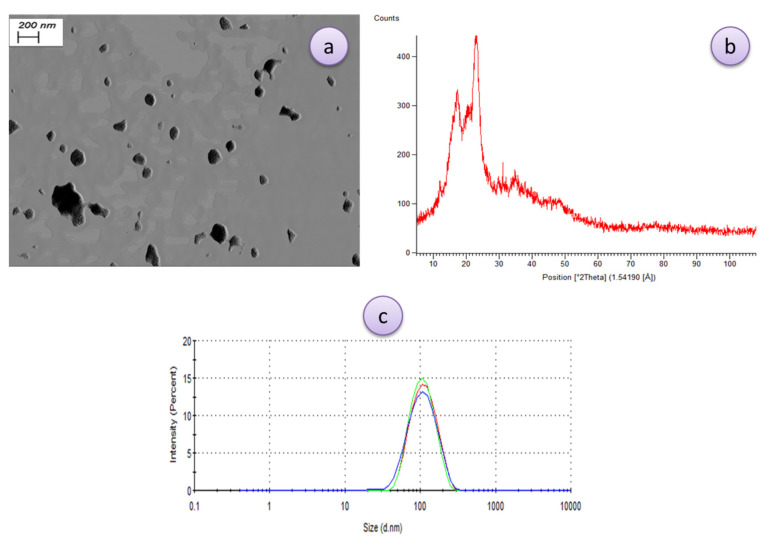
Morphology (**a**), X-ray diffraction (XRD)-pattern (**b**), and size distribution (**c**) characterization of the synthesized starch nanocrystals (SNCs).

**Figure 2 polymers-13-02113-f002:**
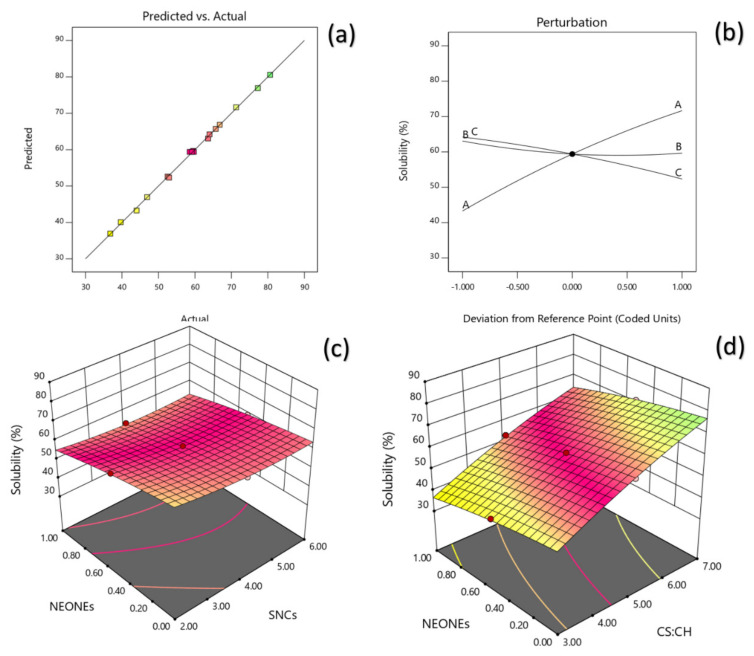
Counter and perturbation graphs (**a**,**b**), the interaction between CS:CH and SNCs concentration and also interaction between SNCs and NEONEs concentration on solubility of the produced film (**c**,**d**).

**Figure 3 polymers-13-02113-f003:**
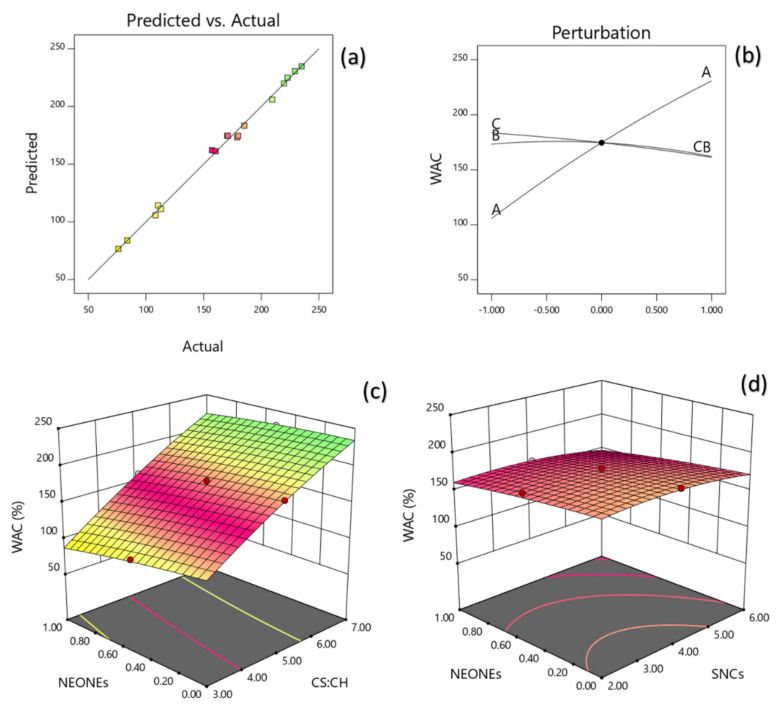
Counter and perturbation graphs (**a**,**b**), the interaction between CS:CH and SNCs concentration and also interaction between SNCs and NEONEs concentration on WAC of the produced film (**c**,**d**).

**Figure 4 polymers-13-02113-f004:**
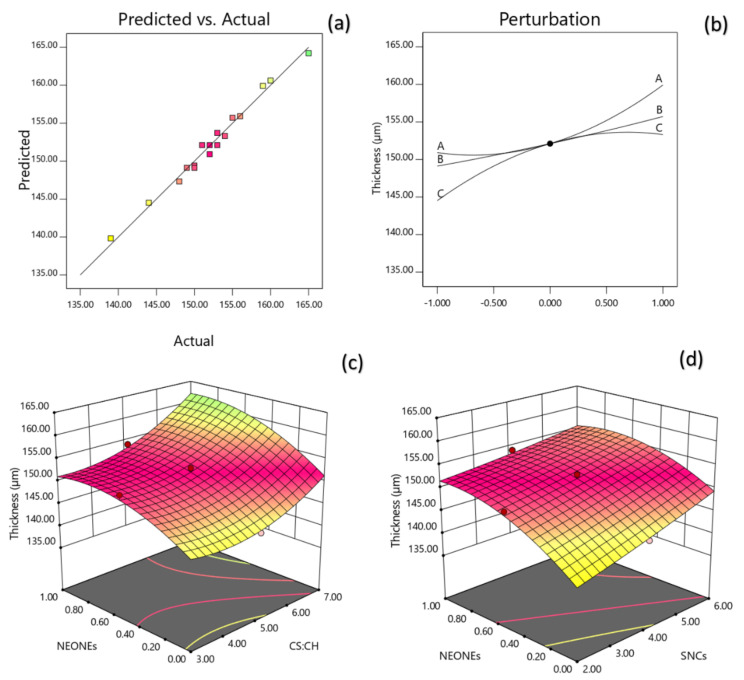
Counter and perturbation graphs (**a**,**b**), the interaction between CS:CH and SNCs concentration and also interaction between SNCs and NEONEs concentration on thickness of the produced film (**c**,**d**).

**Figure 5 polymers-13-02113-f005:**
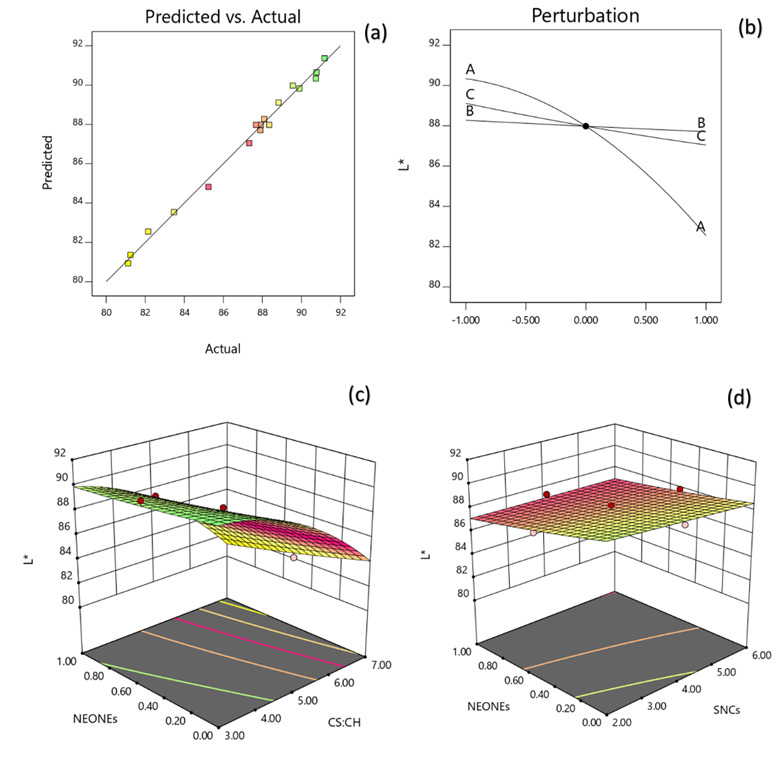
Counter and perturbation graphs (**a**,**b**), the interaction between CS:CH and SNCs concentration and also interaction between SNCs and NEONEs concentration on L* parameter of the produced film (**c**,**d**).

**Figure 6 polymers-13-02113-f006:**
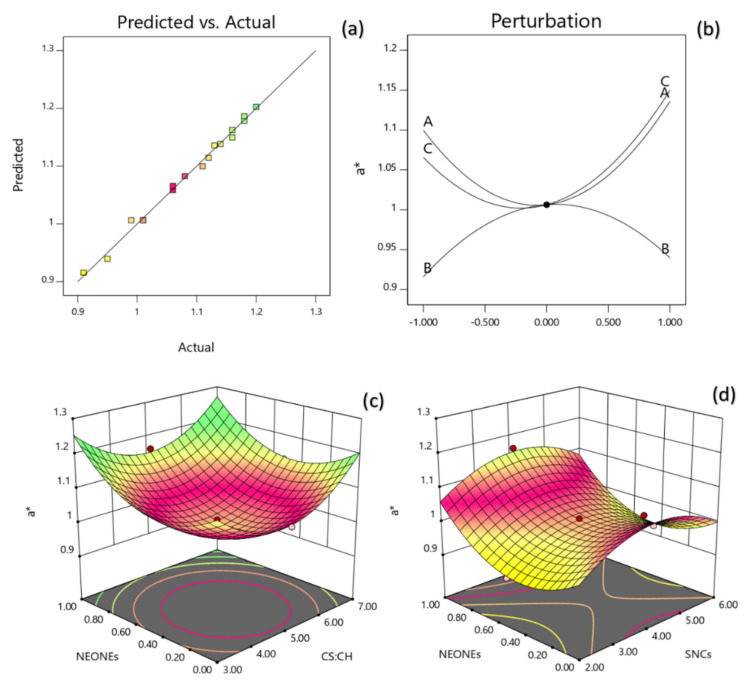
Counter and perturbation graphs (**a**,**b**), the interaction between CS:CH and SNCs concentration and also interaction between SNCs and NEONEs concentration on a* parameter of the produced film (**c**,**d**).

**Figure 7 polymers-13-02113-f007:**
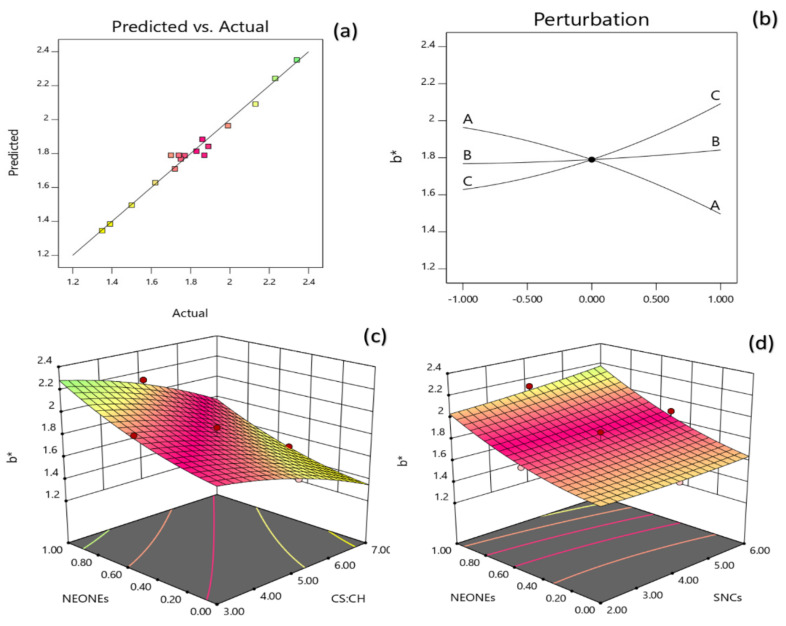
Counter and perturbation graphs (**a**,**b**), the interaction between CS:CH and SNCs concentration and also interaction between SNCs and NEONEs concentration on b* parameter of the produced film (**c**,**d**).

**Figure 8 polymers-13-02113-f008:**
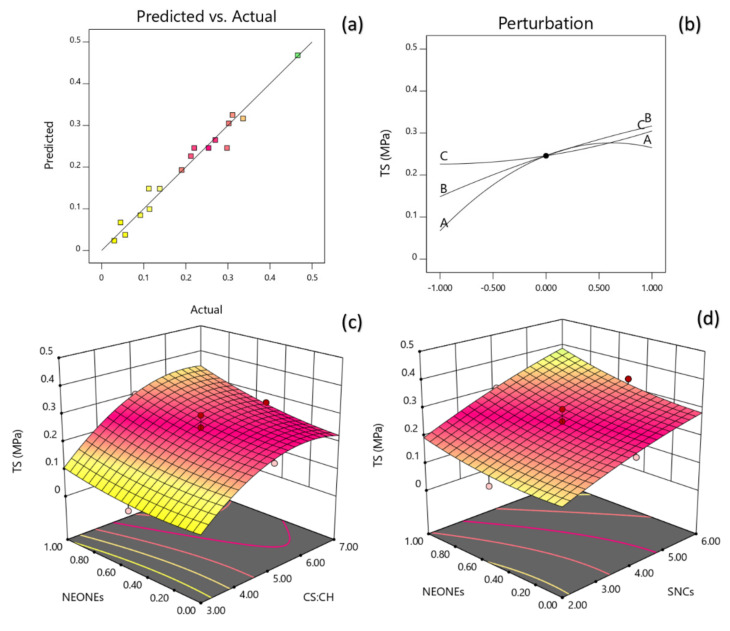
Counter and perturbation graphs (**a**,**b**), the interaction between CS:CH and SNCs concentration and also interaction between SNCs and NEONEs concentration on TS of the produced film (**c**,**d**).

**Figure 9 polymers-13-02113-f009:**
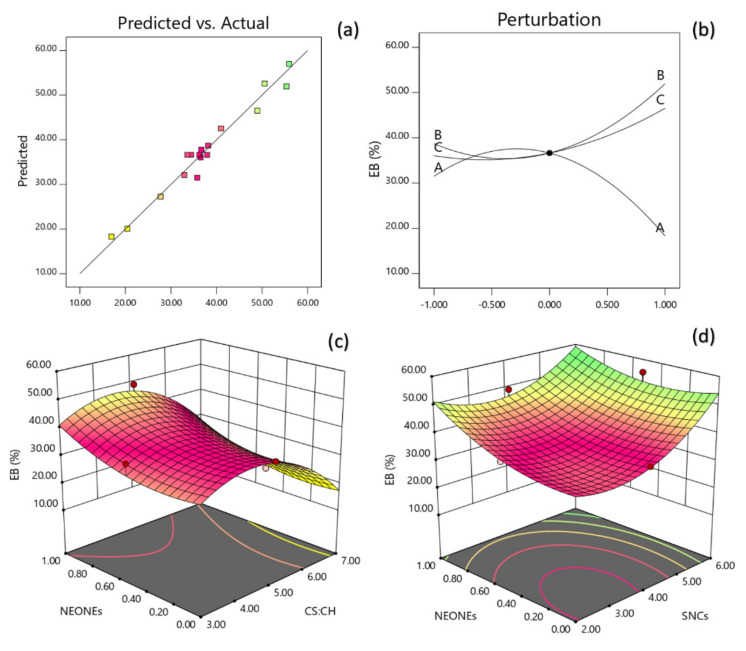
Counter and perturbation graphs (**a**,**b**), the interaction between CS:CH and SNCs concentration and also interaction between SNCs and NEONEs concentration on elongation at break (EB) of the produced film (**c**,**d**).

**Figure 10 polymers-13-02113-f010:**
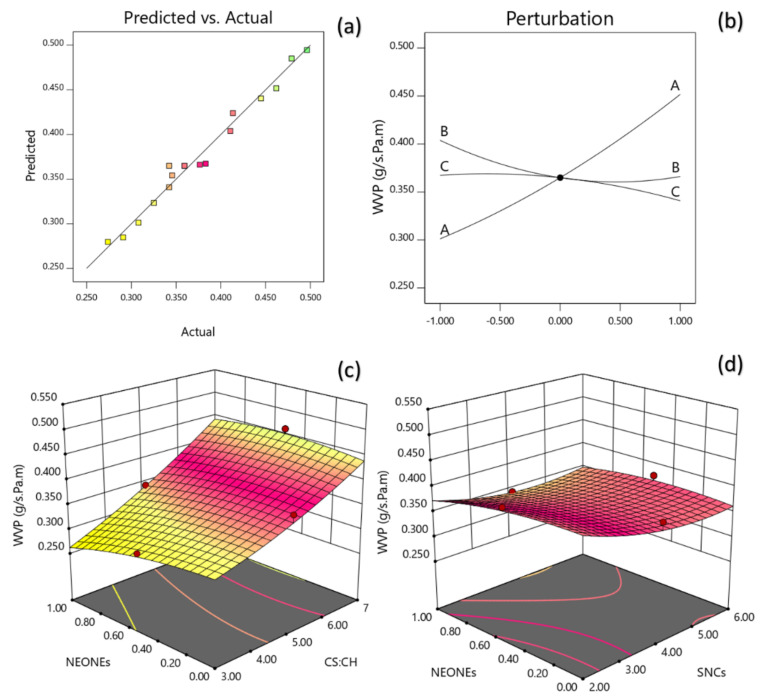
Counter and perturbation graphs (**a**,**b**), the interaction between CS:CH and SNCs concentration and also interaction between SNCs and NEONEs concentration on water vapor permeability (WVP) of the produced film (**c**,**d**).

**Figure 11 polymers-13-02113-f011:**
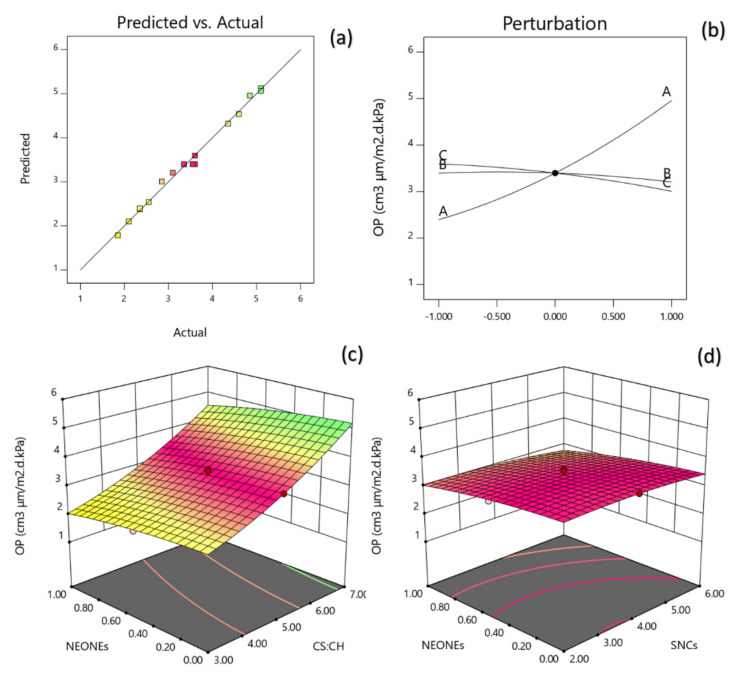
Counter and perturbation graphs (**a**,**b**), the interaction between CS:CH and SNCs concentration and also interaction between SNCs and NEONEs concentration on oxygen permeability (OP) of the produced film (**c**,**d**).

**Table 1 polymers-13-02113-t001:** Experimental design of chitosan–starch nanocomposite films and their corresponding physical and mechanical results.

Run	Independent Variables			Response Variables									
	CS:CH (X_1_, %*w*/*w*)	SNCs (X_2_, %*w*/*w*)	NEONEs (X_3_, %*w*/*w*)	Solubility (Y_1_)	WAC (Y_2_)	TM (Y_3_)	L* (Y_4_)	a* (Y_5_)	b* (Y_6_)	TS (Y_7_)	EB (Y_8_)	WVP (Y_9_)	OP (Y_10_)
1	3.00	6.00	1.00	36.80	76.10	153.00	89.56	1.18	2.34	0.138	55.95	0.274	1.85
2	7.00	4.00	0.50	71.30	229.20	159.00	82.15	1.13	1.50	0.27	16.93	0.462	4.85
3	3.00	2.00	0.00	52.50	110.50	139.00	91.18	1.06	1.83	0.03	27.70	0.345	2.55
4	3.00	4.00	0.50	44.01	108.30	152.00	90.74	1.11	1.99	0.045	35.76	0.308	2.35
5	5.00	6.00	0.50	59.50	157.30	155.00	87.89	0.95	1.89	0.336	55.36	0.376	3.10
6	7.00	6.00	0.00	77.24	219.80	156.00	83.47	1.14	1.39	0.311	32.95	0.413	5.10
7	3.00	6.00	0.00	46.90	112.99	150.00	90.77	1.08	1.77	0.092	50.56	0.325	2.35
8	7.00	2.00	1.00	66.80	222.86	160.00	81.24	1.18	1.72	0.19	37.94	0.479	4.60
9	3.00	2.00	1.00	39.66	83.80	149.00	89.90	1.16	2.23	0.056	40.98	0.291	2.10
10	5.00	4.00	0.50	59.70	170.56	153.00	87.67	0.99	1.70	0.298	33.58	0.342	3.55
11	7.00	6.00	1.00	65.70	209.50	165.00	81.12	1.20	1.86	0.466	36.66	0.445	4.35
12	7.00	2.00	0.00	80.60	235.00	148.00	85.24	1.12	1.35	0.1137	20.43	0.496	5.10
13	5.00	2.00	0.50	63.60	179.14	150.00	88.09	0.91	1.75	0.112	38.17	0.411	3.35
14	5.00	4.00	1.00	52.92	160.51	154.00	87.33	1.16	2.13	0.302	48.95	0.342	2.85
15	5.00	4.00	0.50	59.16	170.97	151.00	88.35	1.01	1.74	0.22	36.23	0.359	3.35
16	5.00	4.00	0.00	64.00	185.30	144.00	88.83	1.06	1.62	0.212	36.50	0.383	3.60
17	5.00	4.00	0.50	58.56	180.01	152.00	87.92	1.01	1.87	0.254	34.41	0.359	3.60

WAC: Water Absorption Capacity, TM: Thickness Measurement, L*: lightness, a*: redness, b*: yellowness, TS: Tensile Strength, EB: Elongation at Break, WVP: Water Vapor Permeability, OP: Oxygen Permeability.

**Table 2 polymers-13-02113-t002:** The phenolic compounds of nettle essential oil identified by gas chromatography mass spectrometry (GC-MS).

Compound	Compound Name	Units	Amount
1	Phenol	ppb	9.9
2	2-chlorophenol	ppb	17.7
3	2,4-dimethyl phenol	ppb	6.8
4	4-chloro-3-methyl phenol	ppb	8.7
5	2,4-dichlorophenol	ppb	47.7
6	2-nitrophenol	ppb	8.1
7	2,4,6-trichlorophenol	ppb	15.3
8	4-nitrophenol	ppb	17.0
9	Pentachlorophenol	ppb	23.5

**Table 3 polymers-13-02113-t003:** Analysis of variance (ANOVA) table for the second-order polynomial models constructed for each of the studied response variables.

Source	DF	Solubility (%)			WAC (%)			TM (µm)		
		MS	F-Value	*p*-Value	MS	F-Value	*p*-Value	MS	F-Value	*p*-Value
Model	9	268.5	752.59	<0.00	4617	195.5	<0.00	63.20	48.70	<0.00
X_1_ (CS:CH)	1	2009	5632.0	<0.00	39021	1652	<0.00	202.5	156.0	<0.00
X_2_ (SNCs)	1	28.97	81.17	<0.00	309	13.10	<0.00	108.9	83.91	<0.00
X_3_ (NEONEs)	1	352.3	987.38	<0.00	1228	52.02	0.00	193.6	149.1	<0.00
X_1_X_2_	1	2.00	5.60	0.04	68	2.89	ns	0.50	0.38	ns
X_1_X_3_	1	0.72	2.02	ns	211	8.97	0.02	8.00	6.16	0.04
X_2_X_3_	1	3.13	8.76	0.02	8.72	0.36	ns	12.50	9.63	/
X_1_^2^	1	9.91	27.76	0.00	110	4.68	ns	29.23	22.52	0.00
X_2_^2^	1	10.42	29.20	0.00	129	5.49	0.05	0.245	0.18	ns
X_3_^2^	1	3.35	9.38	0.01	13.7	0.58	ns	27.39	21.10	0.00
Residual	7	0.35	/	/	23.6	/	/	1.30	/	/
Lack of fit	5	0.36	1.14	0.52 ^ns^	21.6	0.75	0.65 ^ns^	1.42	1.42	0.46 ^ns^
Pure error	2	0.32	/	/	28.5	/	/	1.00	/	/
Core total	16	/	/	/	/	/	/	/	/	/
R^2^	0.99	/	/	/	/	/	/	/	/	/
Adeq precision	95.2	/	/	/	/	/	/	/	/	/
C.V	1.02	/	/	/	/	/	/	/	/	/
R^2^adj	0.99	/	/	/	/	/	/	/	/	/

DF, and MS are degree of freedom, and mean squares, respectively. ns: non-significant. CS: CH represents corn starch: chitosan, SNCs represents starch nanocrystals, NEONEs represents nettle essential oil nanoemulsions, WAC represents water absorption capability, and TM represents thickness measurement.

**Table 4 polymers-13-02113-t004:** ANOVA table for the second-order polynomial models constructed for each the color response variable.

Source	DF	Lightness (L* Value)			Redness (a* Value)			Yellowness (b* Value)		
		MS	F-Value	*p*-Value	MS	F-Value	*p*-Value	MS	F-Value	*p*-Value
Model	9	19.38	109.5	<0.00	0.01	114.7	<0.00	0.01	37.58	<0.00
X_1_ (CS:CH)	1	155.5	856.2	<0.00	0.00	28.11	0.00	0.54	163.6	<0.00
X_2_ (SNCs)	1	0.80	4.56	ns	0.00	12.49	0.009	0.01	4.09	ns
X_3_ (NEONEs)	1	10.69	60.4	<0.00	0.01	153.0	<0.00	0.53	160.8	<0.00
X_1_X_2_	1	0.162	0.91	0.01	0.00	0.001	ns	0.00	0.63	ns
X_1_X_3_	1	1.86	10.5	0.01	0.00	6.94	0.033	0.00	0.631	ns
X_2_X_3_	1	0.36	2.09	ns	0.00	0.001	ns	0.00	2.72	ns
X_1_^2^	1	6.30	33.58	0.00	0.03	288.5	<0.00	0.00	2.88	ns
X_2_^2^	1	0.00	0.00	ns	0.01	143.5	<0.00	0.00	0.18	ns
X_3_^2^	1	0.02	0.15	ns	0.02	239.0	<0.00	0.01	3.92	ns
Residual	7	0.17	/	/	0.00	/	/	0.00	/	/
Lack of fit	5	0.20	1.69	0.41 ^ns^	0.00	0.810	0.63 ^ns^	0.00	0.193	ns
Pure error	2	0.11	/	/	0.00	/	/	0.00	/	/
Core total	16	/	/	/	/	/	/	/	/	/
R^2^	0.99	/	/	/	/	/	/	/	/	/
Adeq precision	32.2	/	/	/	/	/	/	/	/	/
C.V	0.48	/	/	/	/	/	/	/	/	/
R^2^adj	0. 98	/	/	/	/	/	/	/	/	/

DF, and MS are degree of freedom, and mean squares, respectively. ns: non-significant.

**Table 5 polymers-13-02113-t005:** ANOVA table for the second-order polynomial models constructed for each mechanical and barrier response variable.

Source	DF	TS (MPa)			EB (*%*)			WVP (g/s Pa m)		
		MS	F-Value	*p*-Value	MS	F-Value	*p*-Value	MS	F-Value	*p*-Value
Model	9	0.02	25.86	<0.00	196.9	21.48	0.00	0.00	33.96	<0.00
X_1_ (CS:CH)	1	0.09	100.2	<0.00	439.1	47.59	0.00	0.05	273.1	<0.00
X_2_ (SNCs)	1	0.07	72.43	<0.00	438.8	47.88	0.00	0.00	17.17	0.00
X_3_ (NEONEs)	1	0.01	15.83	0.00	273.9	29.88	0.00	0.00	8.48	0.02
X_1_X_2_	1	0.01	13.87	0.00	88.3	9.64	0.01	0.00	3.80	ns
X_1_X_3_	1	0.00	3.25	ns	0.82	0.08	ns	0.00	8.72	0.02
X_2_X_3_	1	0.00	1.25	ns	58.81	6.42	0.03	0.00	1.62	ns
X_1_^2^	1	0.01	17.74	0.00	368.8	40.23	0.00	0.00	1.72	ns
X_2_^2^	1	0.00	0.486	ns	202.3	22.08	0.00	0.00	5.21	0.05
X_3_^2^	1	0.00	1.06	ns	57.93	6.32	0.04	0.00	1.51	ns
Residual	7	0.00	/	/	9.17	/	/	0.00	/	/
Lack of fit	5	0.00	0.4945	0.77 ^ns^	12.1	6.58	0.13 ^ns^	0.00	2.57	0.30 ^ns^
Pure error	2	0.00	/	/	1.84	/	/	0.00	/	/
Core total	16	/	/	/	/	/	/	/	/	/
R^2^	0.97	/	/	/	/	/	/	/	/	/
Adeq precision	18.5	/	/	/	/	/	/	/	/	/
C.V	15.4	/	/	/	/	/	/	/	/	/
R^2^adj	0.93	/	/	/	/	/	/	/	/	/

DF, and MS are degree of freedom, and mean squares, respectively. ns: non-significant.

**Table 6 polymers-13-02113-t006:** ANOVA table for the response surface methodology (RSM) models for oxygen permeability (cm^3^ μm/m^2^ d kPa).

Source	DF	MS	F-Value	*p*-Value
Model	9	1.95	106.03	<0.0001
X_1_ (CS:CH)	1	16.38	889.39	<0.0001
X_2_ (SNCs)	1	0.0902	4.90	ns
X_3_ (NEO-NEs)	1	0.8702	47.24	0.0002
X_1_X_2_	1	0.0050	0.2714	ns
X_1_X_3_	1	0.0113	0.6107	ns
X_2_X_3_	1	0.0113	0.6107	ns
X_1_^2^	1	0.2063	11.20	0.0123
X_2_^2^	1	0.0255	1.38	ns
X_3_^2^	1	0.0255	1.38	ns
Residual	7	0.0184	/	/
Lack of fit	5	0.0188	1.07	0.5469 ^ns^
Pure error	2	0.0175	/	/
Core total	16	/	/	/
R^2^	0.9927	/	/	/
Adeq precision	32.0854	/	/	/
C.V	3.94	/	/	/
R^2^_adj_	0.9661	/	/	/

DF, and MS are degree of freedom, and mean squares, respectively. ns: non-significant.

**Table 7 polymers-13-02113-t007:** Comparison of the experimental and predicted data for the investigated physicochemical properties of produced nanocomposite film.

Value	Nanocomposite Film Physicochemical Attributes				
	WS (%)	WAC (%)	L*	a*	b*
Predicted ^a^	51.56	128.75	89.60	0.96	1.90
Experimental ^b^	52.14 ± 0.23	128.43 ± 2.65	87.63 ± 0.78	0.94 ± 0.03	1.95 ± 0.01
Value	WVP (g/s Pa m)	OP (cm^3^ μm/m^2^ d kPa)	TM (µm)	EB (%)	TS (MPa)
Predicted ^a^	0.335	2.60	154.41	53.54	0.20
Experimental ^b^	0.346 ± 0.001	52.14 ± 0.23	153.21 ± 3.21	52.14 ± 0.33	0.18 ± 0.03

^a^ At optimal point determined by Design Expert software (38:62 of CS:CH, 6.0% of SNC and 0.41% of NEONEs). ^b^ All values are means of triplicates with standard deviation. WS: Water Solubility, WAC: Water Absorption Capacity, L*: lightness, a*: redness, b*: yellowness.

## Data Availability

The data that support the findings of this study are available from the corresponding author.

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
