# Peer review of "Corn Starch-Chitosan Nanocomposite Film Containing Nettle Essential Oil Nanoemulsions and Starch Nanocrystals: Optimization and Characterization"

_polymers, 2021, doi:10.3390/polym13132113_

Round 1

Reviewer 1 Report

The Authors proposed the preparation of the corn starch-chitosan film containing nettle essential oil. The topic is interesting, but there is a lot of data that should be more detailed described and more clear presented. First, show the proves that you made nanostructures of starch – in present form it is not so clear as you state. Some general scheme of how the structure of NPs was performed will help to catch each step of preparation which was made - please add it. Give some graph how your particles look like and combine it with the prepared film structure. For that reason, major revision is needed. Detailed comments are listed below:

Abstract

  • The first sentence is too long, please divide it.
  • in general the abstract is difficult to read - there is total chaos. I.e. just reading the abstract I have no idea what is the reason for use: 2,4-dichlorophenol and pentachlorophenol.
  • please add why do you want to apply this material for packaging, especially, due to antimicrobial properties.

Introduction

  • what does it mean by "prestigious biopolymer"?
  • the use of Nettle should be more explained in the introduction - why did you chose this essential oil, what properties have been studied before, etc.?

Materials and methods

  • "extract" as a separate section "Methods section" - it will be easier to follow up the experiment

Results and discussion

  • fig. 1 please add one more SEM image with higher magnification - add in figure caption how the morphology was studied (SEM?). In the figure caption please detail which method was used. What do orange marks on the top of the graph mean?
  • please explain at which step of the material preparation did you control the particle size?
  • showing the color please add a photo (just from a simple camera) of how the films look like
  • did you make antibacterial activity tests (due to abstract information)? If not please add few comments in the results and discussion (use some references if needed)
  • enlarge the fonts and make them more visible on the graphs

Conclusions

- summarize the results of water and gas permeability tests

Author Response

Dear Reviewer,

Thanks for your precise and valuable revision. Your comments are addressed in the text and summarized as below:

Abstract

  • The first sentence is too long, please divide it.

The sentence is corrected.

  • in general the abstract is difficult to read - there is total chaos. I.e. just reading the abstract I have no idea what is the reason for use: 2,4-dichlorophenol and pentachlorophenol.

The sentence about major bioactives of NEO is deleted, thanks for your comment.

  • please add why do you want to apply this material for packaging, especially, due to antimicrobial properties.

The reason is added to the abstract.

Introduction

  • what does it mean by "prestigious biopolymer"?

We meant valuable, corrected.

  • the use of Nettle should be more explained in the introduction - why did you chose this essential oil, what properties have been studied before, etc.?

The antimicrobial and antioxidant attributes are regarded as most important factors for food packaging applications.

Materials and methods

  • "extract" as a separate section "Methods section" - it will be easier to follow up the experiment

As you suggested, the Reagents and Methods separated in two sections.

Results and discussion

  • fig. 1 please add one more SEM image with higher magnification - add in figure caption how the morphology was studied (SEM?). In the figure caption please detail which method was used. What do orange marks on the top of the graph mean?

A new micrograph is taken and added instead, and the orange marks are also removed.

  • please explain at which step of the material preparation did you control the particle size?

After the synthesis of nanoparticles.

  • showing the color please add a photo (just from a simple camera) of how the films look like
  • did you make antibacterial activity tests (due to abstract information)? If not please add few comments in the results and discussion (use some references if needed)

Not actually, a relevant sentence is added to the conclusion section.

  • enlarge the fonts and make them more visible on the graphs

A better quality of fonts is included.

Conclusions

- summarize the results of water and gas permeability tests

The improvement of gas and water properties is currently mentioned in the conclusion section.

Reviewer 2 Report

This work will add to the knowledge of corn starch-chitosan nanocomposite film containing nettle essential oil and starch nanocrystals once a deep revision was carried out.

The topic of the paper is interesting, and authors have performed a well-planned and executed work. However, Materials & Methods section as well as Results & Discussion sections should be revised, especially the informed results regarding WVP.

Specific comments are included in the attached file.

Author Response

Dear Reviewer,

Thanks for your precise and valuable revision. Your comments are addressed in the text.

Round 2

Reviewer 1 Report

The Authors mostly improved the text. Nevertheless, still, major revision is needed before the presentation. See details here:

Please use the mdpi template.

DLS measurements - please add the scattering angle.

I can not agree that the particles were uniformed. Looking at SEM images, there are various too low and should be enhanced before publication. What is more, the discussion of the XRD is poor. It is obvious that amorphous structures will not "generate" peaks. Pease focuses on the results and describes what was detected during XRD examinations.

What is more, you can not count only on the scattered light intensity in the DLS measurements (for details see here: Nanoparticles Size Determination by Dynamic Light Scattering in Real (Non-standard) Conditions Regulators—Design, Tests and Applications. practical Aspects of Chemical Engineering
or here Molecules2020,25, 2696.

As I mentioned before, please add the graph where step by step the material preparation will be explained.

Author Response

Responses to Reviewer’s comment:

We wish to thank all reviewers for re-evaluating the revised manuscript and providing us with additional feedbacks which enabled us to generate our work at the highest possible quality. Herein we document the changes we have made in response to the additional comments made by reviewers. We believe that this version of paper is suitable for publication.

The Authors mostly improved the text. Nevertheless, still, major revision is needed before the presentation. See details here:

Please use the mdpi template.

Response: We have revised the text based on the mdpi template.

DLS measurements - please add the scattering angle.

Response: We gladly accept this suggestion. The scattering angle was included in the revised paper.

I can not agree that the particles were uniformed. Looking at SEM images, there are various too low and should be enhanced before publication. What is more, the discussion of the XRD is poor. It is obvious that amorphous structures will not "generate" peaks. Please focuses on the results and describes what was detected during XRD examinations.

Response: Thank you for this insightful comment. We have revised the text according to the reviewer’s suggestion. Given the small size of the particles (100 -200 nm), it would be challenging to adjust the contrast/brightness of the SEM image under this magnification. This could be the reason that image looks blurry.

We have modified the XRD section and provided several new statements following the reviewer’s suggestion.

What is more, you can not count only on the scattered light intensity in the DLS measurements (for details see here: Nanoparticles Size Determination by Dynamic Light Scattering in Real (Non-standard) Conditions Regulators—Design, Tests and Applications. practical Aspects of Chemical Engineering
or here Molecules2020, 25, 2696.

Response: We agree with the reviewer. We decided to present droplet size distribution by intensity because the data of droplet size distribution by number and volume were similar to those of intensity. Therefore, we believe one set of data should be enough for the paper.

We found the reference provided by reviewer very interesting and expanded the discussion of the relevant section on DLS.

As I mentioned before, please add the graph where step by step the material preparation will be explained.

Response: Thank you for this suggestion. We are limited with number of the figures/tables and the paper does not allow us to include additional graph. So, although, the reviewer’s suggestion is appropriate, we respectfully preferred to not include it.

Round 3

Reviewer 1 Report

The Authors mostly improved the manuscript due to the Reviewers suggestions. The paper is almost read to presentation in the Polymers.
Please carefully check the quality of the SEM image - now it is not enough.